# Metabolomics facilitates differential diagnosis in common inherited retinal degenerations by exploring their profiles of serum metabolites

Wei-Chieh Wang [1,12], Chu-Hsuan Huang [2,3,12], Hsin-Hsiang Chung [1], Pei-Lung Chen [4,5], Fung-Rong Hu [6,7], Chang-Hao Yang [6,7], Chung-May Yang [6,7], Chao-Wen Lin [6], Cheng-Chih Hsu [1,8,13] ✉ & Ta-Ching Chen [6,9,10,11,13] ✉

The diagnosis of inherited retinal degeneration (IRD) is challenging owing to its phenotypic and genotypic complexity. Clinical information is important before a genetic diagnosis is made. Metabolomics studies the entire picture of bioproducts, which are determined using genetic codes and biological reactions. We demonstrated that the common diagnoses of IRD, including retinitis pigmentosa (RP), cone-rod dystrophy (CRD), Stargardt disease (STGD), and Bietti's crystalline dystrophy (BCD), could be differentiated based on their metabolite heatmaps. Hundreds of metabolites were identified in the volcano plot compared with that of the control group in every IRD except BCD, considered as potential diagnosing markers. The phenotypes of CRD and STGD overlapped but could be differentiated by their metabolomic features with the assistance of a machine learning model with 100% accuracy. Moreover, *EYS*-, *USH2A*-associated, and other RP, sharing considerable similar characteristics in clinical findings, could also be diagnosed using the machine learning model with 85.7% accuracy. Further study would be needed to validate the results in an external dataset. By incorporating mass spectrometry and machine learning, a metabolomics-based diagnostic workflow for the clinical and molecular diagnoses of IRD was proposed in our study.

Inherited retinal degeneration (IRD) contains a group of retinopathies characterized by high heterogeneity in phenotypes and a widely variable genetic background. The incidence of IRD is ~1 in 2000 individuals worldwide and is attributed to a significant proportion of people with profound visual impairment or blindness[1]. Phenotypes of IRD are diverse, and overlapping in clinical presentation is common[2,3]. Nowadays, over 300 genes have been considered causative for IRDs and were reported on RetNet (https://sph.uth.edu/retnet/). Not only multiple genes may result in one IRD, such as retinitis pigmentosa (RP) and Leber congenital amaurosis (LCA), but many genes may also be associated with different IRDs. It is even more difficult to predict the causative genes in patients with similar phenotypes in these diseases, despite some genotype-phenotype correlations being proposed[4–6]. This complexity makes the diagnosis of IRDs very challenging.

Owing to the development of molecular diagnosis, for example, next-generation sequencing (NGS) technology, the genetic spectrum and diagnostic accuracy of IRDs have significantly progressed in this decade[7]. For example, the Taiwan IRD Project (TIP) platform achieved a 57.1% detection rate in identifying causative genes by incorporating panel-based NGS technology and could be further higher with whole-

exome sequencing (WES) and whole-genome sequencing (WGS)[8]. However, it is still time-consuming and expensive to check the comprehensive genome-wide profile of every potential patient and related family member because of the wide genetic spectrum of IRD, not to mention nationwide screening[9]. To promote public health and early detection of potential patients with IRD, practical pre-test clinical information may be beneficial[10–12].

Metabolomics, the emerging tool investigating comprehensive small molecules (<1500 Da) constitution of biological samples, provides instantaneous phenotypic information as metabolites reflect both genetic and environmental factors[13]. Mass spectrometry (MS) is one of the most widely used platforms for metabolomic analysis due to its high sensitivity and selectivity[14]. Therefore, MS-based metabolomics analysis has been widely applied to medical science and clinical research, in which the health status of participants is obtained using molecular fingerprints[15,16]. MS-based metabolomics approach has also shown great potential in ophthalmologic diseases research, such as glaucoma[17], diabetic retinopathy[18], age-related macular degeneration (AMD)[19,20], and dry eye[21]. It reveals potential biomarkers that improve diagnostic accuracy and possibly introduce available treatment options for eye diseases[22–24]. The metabolic profiling may not only reveal potential biomarkers that improve diagnostic accuracy but also possibly introduce available treatment options, which would benefit the clinical aspect of IRDs.

High-throughput serum metabolic analyses and accurate compound identification can be achieved using liquid chromatography-high-resolution tandem mass spectrometry (LC-HR-MS/MS). The machine learning algorithm enabled the extraction of potential biomarkers associated with specific IRD subtypes from complex MS datasets, providing excellent performance in distinguishing different phenotypes or genotypes of these heterogeneous diseases.

In this study, we aimed to establish a model based on serum metabolomic profiles and machine learning for classifying specific IRDs before conducting genetic tests. Based on the database from the TIP, RP was the most common diagnosis, accounting for 66% of the cases, followed by macular dystrophy, CD/CRD, and BCD, accounting for 12.5%, 6.1%, and 3.5%, respectively[8]. The most prevalent causative genes in RP probands in this study were *USH2A, EYS, PRPF31*, and *ABCA4*. Moreover, the most common diagnosis in the macular dystrophy group is STGD[8]. This study focuses on these IRD subtypes with the highest prevalence in Taiwan.

## Results

### Metabolomic features of IRDs revealed using heatmap and selected metabolites

A total of 155 participants were enrolled in the present study, including 70 diagnosed with RP, 20 with STGD, 21 with CD/CRD, 16 with BCD, and 28 healthy participants in the control group (Fig. 1); the average age was $48.7 \pm 14.3$, $29.0 \pm 21.1$, $41.4 \pm 14.4$, $49.2 \pm 13.3$, and $43.4 \pm 17.3$-year-old in each group, respectively. The participants in the STGD group were significantly younger than those in the control, BCD, and RP groups [analysis of variance (ANOVA) with Tukey's honestly significant difference test, $P = 0.024$, 0.003, and <0.001, respectively]. There was no statistical difference between IRD groups and healthy control for the presence of systemic disease, body mass index (BMI), and habit of smoking ($P = 0.953$, 0.895, and 0.726, respectively). The prevalence of anti-oxidant supplementation was similar in different IRD groups as well ($P = 0.709$) (Supplementary Table 1).

A heatmap of the serum metabolic profiles of all the participants is shown in Fig. 2a. Among the metabolites evaluated, we selected 40 metabolites with the most significant differences among the groups using ANOVA. There are distinct characteristics and patterns in patients with different IRD subtypes. For example, elevated D-xylonate, citronellyl acetate, and hexadecanedioic acid levels were observed in all IRD subtypes, except BCD, compared with that of the control group (Fig. 2b–d). In contrast, decreased concentrations of N-undecanoylglycine and the other three glycerophospholipids, phosphatidylserine (14:1/16:0), phosphatidylcholine [16:0/9:0(CHO)], and phosphatidylcholine (19:1) were

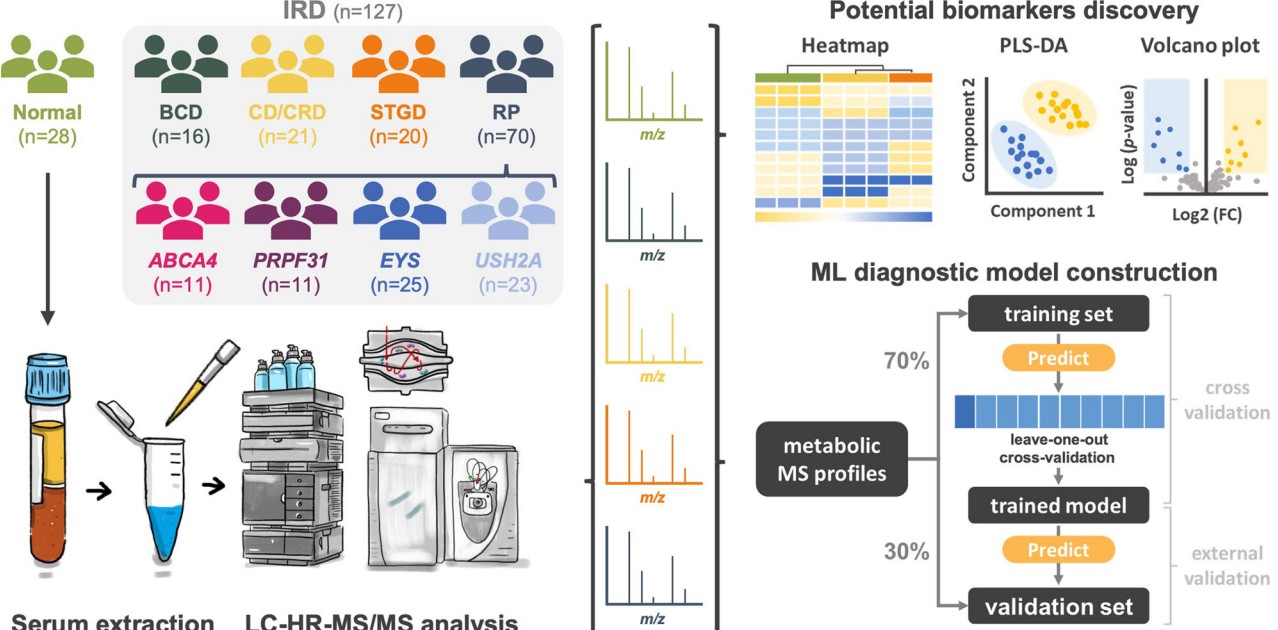

**Fig. 1 | Flow chart of the present study.** Metabolomic analysis using MS was performed in enrolled IRD cases and the control groups. The heatmap, PLS-DA, and volcano plots were used to reveal the difference in metabolomic profile between IRDs and the control group. By incorporating metabolomic information, a machine learning model was constructed for differentiating specific IRDs that cannot be distinguished by clinical examination. MS, mass spectrometry, IRD, inherited retinal degeneration, PLS-DA, partial least squares-discriminant analysis, BCD, Bietti's crystalline dystrophy, CRD, cone-rod dystrophy, RP, retinitis pigmentosa, STGD, Stargardt disease; LC-HR-MS/MS, liquid chromatography-high-resolution tandem mass spectrometry, ML, machine learning.

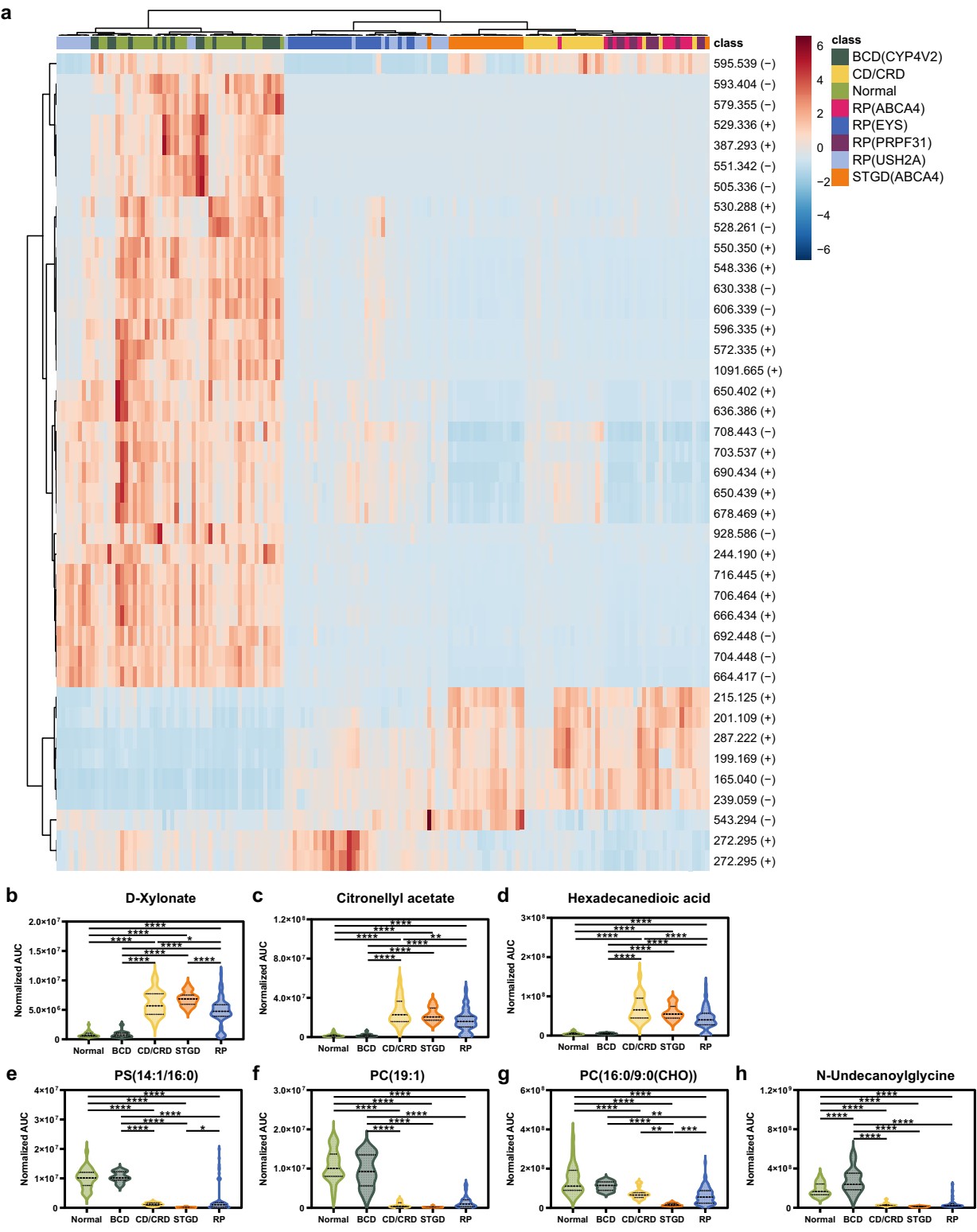

found in all IRD subtypes, except BCD, compared with that of the control group (Fig. 2e–h). Therefore, a great similarity was found between the control and BCD groups, while other subtypes of IRD exhibited different patterns of metabolites compared with that of the control group. The levels of these identified metabolites after adjustment with age are shown in Supplementary Fig. 1, respectively.

**PLS-DA plot in each IRD revealed diagnostic potential**

Based on fundus appearance and clinical pathophysiology, IRD could be preliminarily divided into three groups: cone-predominant degenerative diseases (e.g., CD/CRD and STGD), rod-predominant degenerative diseases (e.g., RP), and crystalline-deposition retinopathy (e.g., BCD). PLS-DA score and volcano plots are shown in Fig. 3. The metabolomic profiles of RP, STGD, and CD/CRD could be successfully

**Fig. 2 | Metabolomics profile and selected identified metabolites in metabolomics analysis in different IRD groups. a** The heatmap of 40 identified metabolites with the most significant differences among the seven IRD subgroups and control group were selected using ANOVA. The red color represents the upregulated metabolites, while the blue color represents the downregulated metabolites in each enrolled case. For (**2b–h**), statistical analysis was performed using analysis of variance (ANOVA) with Tukey's honestly significant difference test (two-sided). Results are indicated by: Nonsignificant; (ns), $p > 0.05$; *$p \leq 0.05$; **$p \leq 0.01$; ***$p \leq 0.001$; ****$p \leq 0.0001$. The levels of D-xylonate (**b**), citronellyl acetate (**c**), and hexadecanedioic acid (**d**) were higher in the CD/CRD, STGD, and RP groups than those in the BCD and control groups. The levels of phosphatidylserine (14:1/16:0) (**e**), phosphatidylcholine (19:1) (**f**), phosphatidylcholine [16:0/9:0(CHO)] (**g**), and N-undecanoylglycine (**h**) were lower in the CD/CRD, STGD, and RP groups than those in the BCD and control groups. Source data are provided as a Source Data file. IRD, inherited retinal degeneration; BCD, Bietti's crystalline dystrophy, CRD, cone-rod dystrophy, RP, retinitis pigmentosa, STGD, Stargardt disease, ANOVA, analysis of variance, PC, phosphatidylcholine, PS, phosphatidylserine.

distinguished from healthy participants in the PLS-DA score plots, whereas BCD largely overlapped with healthy participants. Further analyses revealed that patients with different genotypes tended to have different metabolic conditions despite having similar phenotypes. As shown in Fig. 3a, patients with RP had different metabolic profiles than healthy participants. In further subtype analysis, in Fig. 3b, d, f, h, RP with different disease-causing genes tended to have different metabolic expressions. Different RP genotypes could be better separated from healthy participants in the PLS-DA score plot than pooling them together as an overall RP group, suggesting that different metabolic features exist between different RP phenotypes.

In contrast, patients with the same disease-causing gene but different clinical subtypes share similar metabolic features. For example, as shown in Fig. 3j, patients with STGD and RP with the same genetic mutation, *ABCA4*, exhibited similar metabolic profiles in the PLS-DA score plot.

## Volcano plots quantified the difference between the IRD and control groups

The volcano plots in Fig. 3a–h revealed quantifying the differences in metabolic composition between IRDs and healthy participants corresponding to each PLS-DA score plot. The criteria for significant features highlighted in plots were defined as a false discovery rate <0.05 based on the Benjamini–Hochberg test and fold change >2. While 147–260 features were identified in other IRDs when compared with healthy participants, no significantly different metabolite was found between BCD and healthy participants. The metabolic compositions of *ABCA4*-associated RP and STGD were similar, and only 19 features were identified (Fig. 3j). The Volcano plots after adjustment with age were shown in Supplementary Fig. 2, revealing completely identical trends mentioned above. The number of metabolites with significant differences between each group is shown in Supplementary Table 2. Lipid metabolites contributed to a major part of the differences in each group, especially in *EYS*- and *USH2A*-associated RP.

## Propose machine learning models in rod- and cone-predominant IRDs

As described above, IRD could be preliminarily divided into the following three groups: cone-predominant degenerative diseases, rod-predominant degenerative diseases, and crystalline-deposition retinopathy. Cone-predominant degenerative diseases such as CD/CRD and STGD could present similar clinical manifestations. However, by analyzing the metabolites shown in Fig. 3i, we could better differentiate the two groups without a genetic diagnosis. Similar results have been observed in rod-predominant degenerative diseases. Different genotypes of RP are often indistinguishable clinically but could be separated through metabolite studies, as shown in Fig. 3. Therefore, we attempted to establish a machine learning model to provide first-line diagnostic ability by incorporating big data from metabolomic information.

The first model was established for the diagnosis of CD/CRD, STGD, and healthy participants. As shown in Fig. 4b, both the training and validation sets achieved 100% sensitivity and specificity. The accuracy for the training set based on the leave-one-out cross-validation is 100 ± 0%. Five metabolites were selected as diagnostic features using a machine learning model: dodecanamide,

hexadecanedioate, N-undecanoylglycine, diacylglycerol, and N8-acetylspermidine (Supplementary Fig. 3a and Supplementary Table 3). The second model was established to predict *EYS*- and *USH2A*-associated RP, which are the most prevalent subtypes according to the TIP database. The training and validation sets achieved 83.7% (84 ± 16% in the leave-one-out cross-validation) and 85.7% accuracy, respectively (Fig. 4d). Fourteen metabolites were selected as diagnostic features in this model (Supplementary Fig. 3b and Supplementary Table 3); half (7 out of 14) belonged to the glycerolipid class, and 4 were polyunsaturated fatty acids. According to this finding, the targeted machine-learning model could demonstrate value in differentiating between cone- and rod-predominant IRDs. Therefore, by incorporating the thinking process from clinical information into the targeted machine learning model in metabolomic analysis, we propose a diagnostic flowchart to facilitate the diagnosis of IRD when approaching the genetic confirmation test (Fig. 5b).

## Discussion

Owing to the wide variety of genetic and phenotypic spectra of IRD, precise diagnosis for each individual and family member has been a challenge in clinical practice[25]. With advances in NGS-based molecular diagnosis, the diagnostic rate and genetic spectrum of IRD have been broadened even further. To promote more time- and cost-effective strategies for genetic testing, phenotype-guided, population-based, and clinically directed tiered approaches have been proposed[10–12,26,27]. In the present study, we provide an alternative approach by investigating the information obtained from the metabolomic analysis.

As an end product of systemic metabolism, the profiling of metabolites can serve as a bridge between genetic transcription and the physiological environment. Metabolomics studies have been introduced in ophthalmologic studies in several fields[24,28–30]. For example, Lains et al. found that metabolites involved in the glycerophospholipid pathway were strongly correlated with progress in age-related macular degeneration[20]. Vehof et al. proved that the decrease in serum androgens was associated with the development of dry eye disease[21]. Zuo et al. found a link between specific metabolites and the presence of diabetic retinopathy by establishing a machine-learning model[31]. In the present study, we explored the metabolomic profiles in common subtypes of IRD, including RP, STGD, CD/CRD, and BCD, to determine if there are significant differences among groups and compared them to the healthy population. The results showed that the serum metabolomic profile differed significantly between healthy participants and those with IRD. The profiles also varied widely among the different IRD subtypes. Although they presented with a phenotype similar to that of RP, patients with different genotypes also exhibited diverse serum metabolomic profiles. These results inspired us to incorporate metabolomic studies for disease screening and diagnosis when approaching patients suspected of having IRD before conducting genetic consultation.

One important finding of our study is that the metabolomic profile is more closely related to genetic variations than to the phenotypic characteristics of the fundus. For example, *EYS* and *USH2A* are both important and common disease-causing genes in autosomal recessive RP, whereas patients in these two subgroups may have different metabolomic profiles. In contrast, the metabolomic profile was

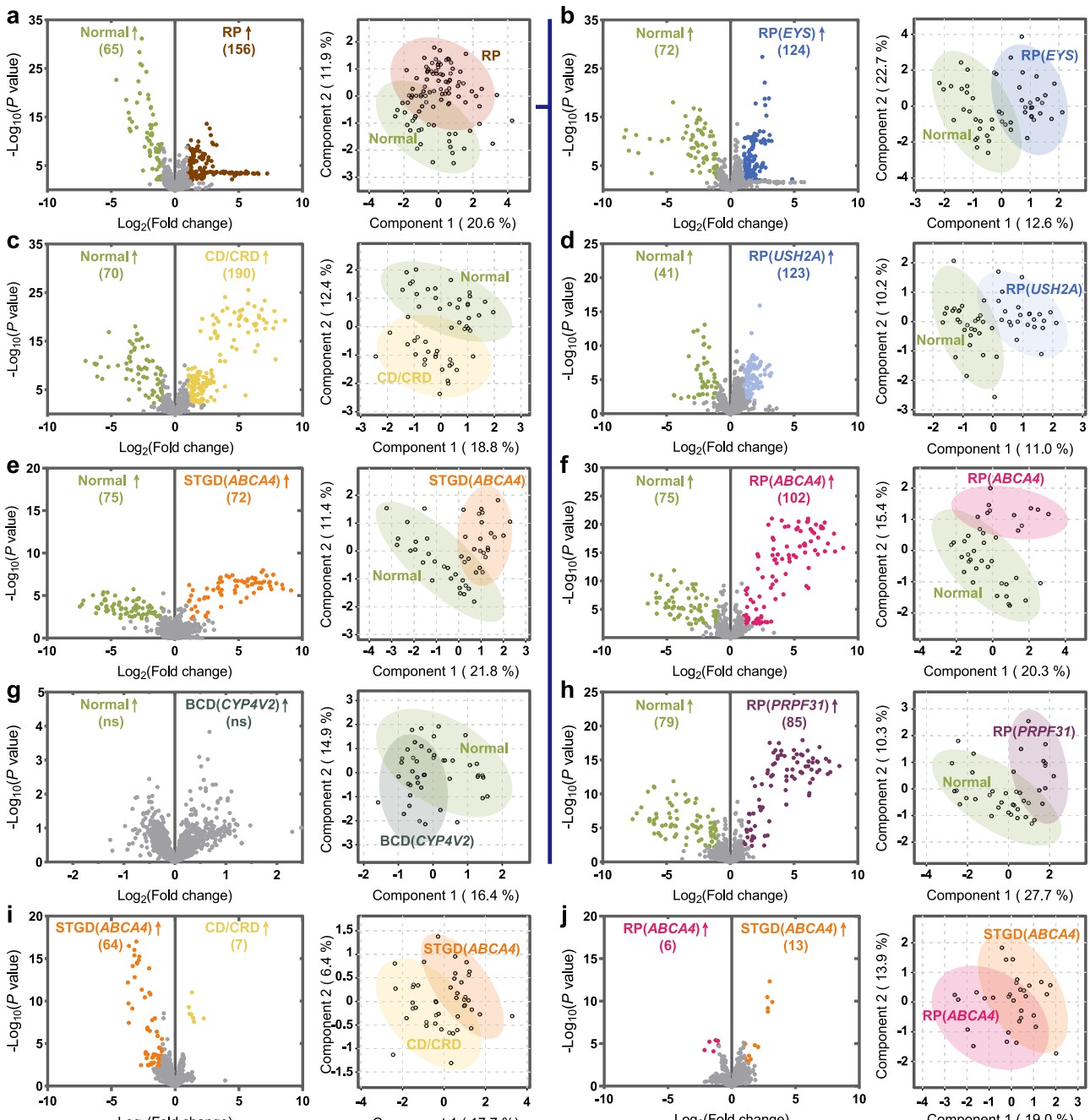

**Fig. 3 | PLS-DA and volcano plots of each IRD subgroup compared with that of the control group.** The significant features highlighted in the volcano plot were defined as having a false discovery rate <0.05 (Benjamini–Hochberg test, two-sided) and fold change >2. The number of metabolite features with significant differences between each IRD subgroup and control group was shown in the parentheses in each volcano plot. No metabolite feature could be identified between the BCD and control group (**g**). The PLS-DA plot showed that the RP, CD/CRD, and STGD groups could be distinguished from the control group in metabolomic analysis (**a**, **c**, **e**), while the overlapping area in the BCD group was more prominent (**g**). Moreover, when we further sub-grouped RP by genotype, including *EYS*, *USH2A*, *ABCA4*, and *PRPF31*, the metabolomic analysis showed more distinguishable results in the PLS-DA plot (**b**, **d**, **f**, **h**). **i** Compared between CD/CRD and STGD, 71 features could be identified with significant differences from the volcano plots, and the two groups could be distinguished in the PLS-DA plot. **j** Compared with RP with the ABCA genotype, STGD showed that only 19 features in the volcano plots could be identified, and the overlapping of the PLS-DA plot was prominent. Despite the significant phenotypic differences, RP and STGD with the same genotype, *ABCA4*, showed similar metabolomic profiles. Source data are provided as a Source Data file. IRD, inherited retinal degeneration, BCD, Bietti's crystalline dystrophy, CRD, cone-rod dystrophy, RP, retinitis pigmentosa; STGD, Stargardt disease, PLS-DA, partial least squares-discriminant analysis.

relatively similar between *ABCA4*-associated STGD and RP despite their distinct phenotypes. These results indicate that different disease-causing genes have different pathological mechanisms in retinal degeneration. Notably, some studies have shown that genotypes predispose the disease progression and poor outcomes for patients with IRD[3,6,25,32,33].

Another interesting finding is that, unlike other subtypes of IRD, we found that the metabolomic profile of patients with BCD is highly similar to that of healthy participants and apart from other subtypes of IRD. No metabolites were found to be significantly different between BCD and healthy participants. BCD is more prevalent in East Asia and is caused by mutations in the *CYP4V2* gene, which is responsible for the

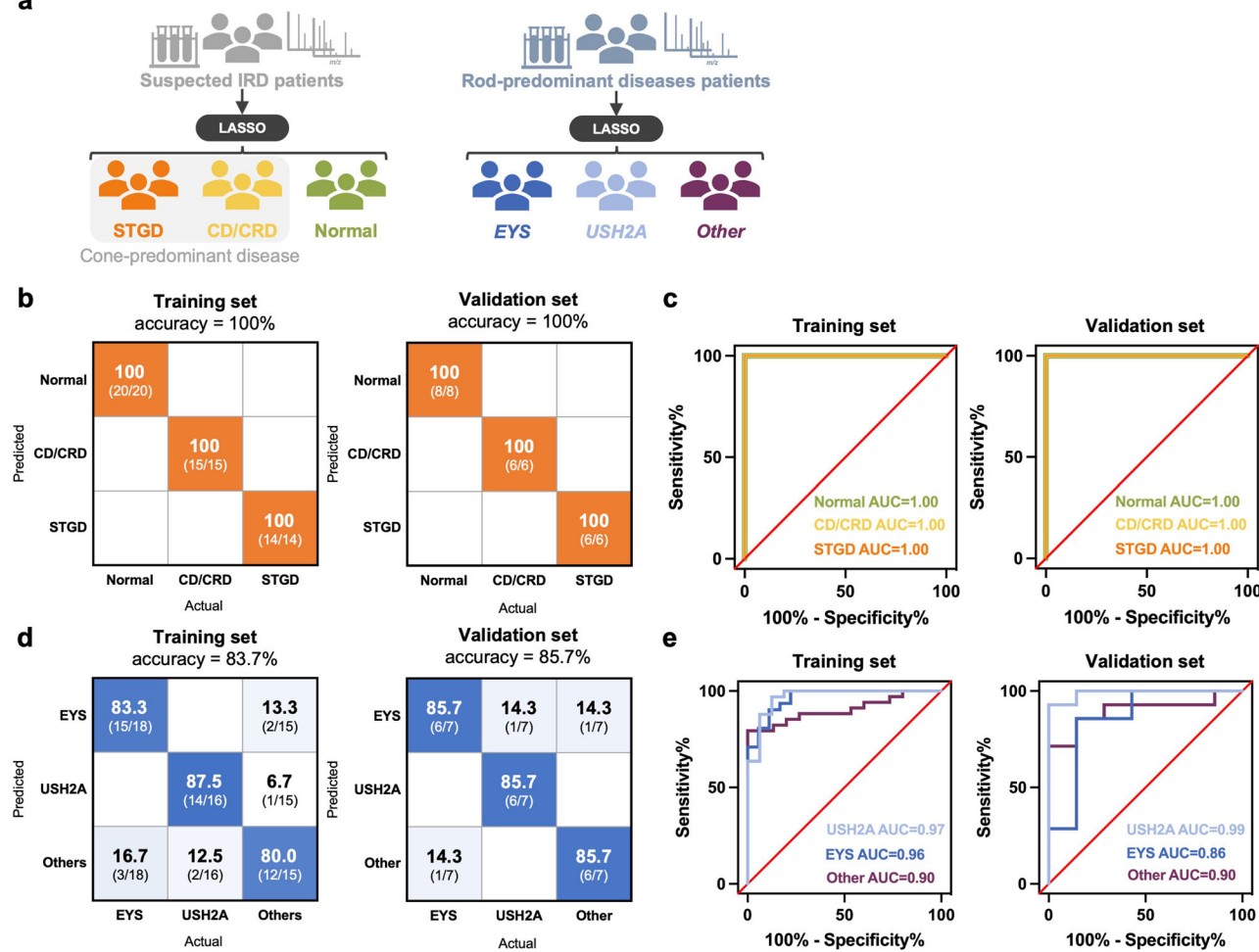

**Fig. 4 | Performance of the machine-learning LASSO model for IRD subtypes classification. a** Two diagnostic models were established to differentiate (1) CD/CRD, STGD, and control group and (2) RP with *EYS*, *USH2A*, and other genotypes, using the machine learning LASSO model. **b** The sensitivity and specificity were both 100% in the training and validation sets of the cone-predominant disease diagnosis model. **c** The area-under-curve (AUC) was 1.0 in the three subgroups in the training and validation set. **d** The diagnostic accuracy was 83.7% in the training set and 85.7% in the validation set of the RP diagnosis model. **e** The AUC of the RP diagnosis model for the *USH2A*, *EYS*, and other genotypes in the training and validation set. Source data are provided as a Source Data file. LASSO, Least Absolute Shrinkage and Selection Operator, IRD, inherited retinal degeneration, CRD, cone-rod dystrophy, RP, retinitis pigmentosa, STGD, Stargardt disease, AUC, area-under-curve.

transformation of polyunsaturated fatty acids[34]. Therefore, the pathogenesis of BCD is primarily a dysfunction of lipid metabolism, and the mechanism is much different from other subtypes of IRD, which generally affect enzymes involved in the visual pathway or protein, maintaining structural stability in photoreceptors and retinal pigmented epithelium cells.

Specific metabolites could be identified from the heatmap and volcano plots, revealing different pathogens in different IRD subtypes and serving as potential biomarkers. From the heatmap, a higher concentration of xylonate and hexadecanedioic acid has been found in the serum of patients with IRD except for those with BCD, both participating in an oxidative reaction as the role of precursor and end product[35,36]. The increased concentration of these metabolites indicated higher oxidative stress in the IRD disease process, which could be associated with retinal cell apoptosis[37]. In contrast, lower concentrations of three different glycerophospholipids, phosphatidylserine (14:1/16:0), phosphatidylcholine [16:0/9:0(CHO)], and phosphatidylcholine (19:1) were observed in IRD samples excluding the BCD. Glycerophospholipids are crucial components of photoreceptor cells, and a reduction of glycerophospholipids in the serum has been reported in patients with age-related macular degeneration[20], which shares similar features of photoreceptor impairment with IRDs.

Furthermore, the highest serum level of N-undecanoylglycine was found in BCD, which was also elevated in patients with various fatty acid oxidation disorders[38]. The causative gene of BCD, CYP4V2, participates in multiple steps of fatty acid oxidation[39]. From the large amounts of metabolites identified in the volcano plot, several metabolites showed correlations with different IRDs. For example, the highest all-trans retinal levels were observed in the CD/CRD group, followed by the RP group. This finding corresponds to a greater disturbance in retinal homeostasis and more severe photoreceptor impairment in CD/CRD and RP in IRDs. Moreover, higher serum levels of 5-hydroxyeicosatetraenoic acid, 5-oxo-eicosatetraenoic acid, leukotriene A4, and leukotriene B4 were found in the CD/CRD group, and these metabolites participated in the inflammatory response during oxidative stress, implying that interference of inflammatory regulation might be involved in the pathogenesis of CD/CRD[40]. The change in the abundance of serum metabolites of patients with IRD corroborates the hypothesis that metabolic profiling is associated with disease phenotypes, which paved the way for IRD subtype diagnosis and gained insight into metabolic pathways that may be related to these diseases.

Artificial Intelligence has been increasingly used in clinical medicine. Many studies have attempted to apply machine learning technologies to help diagnose retinal disorders, including IRD, mainly in

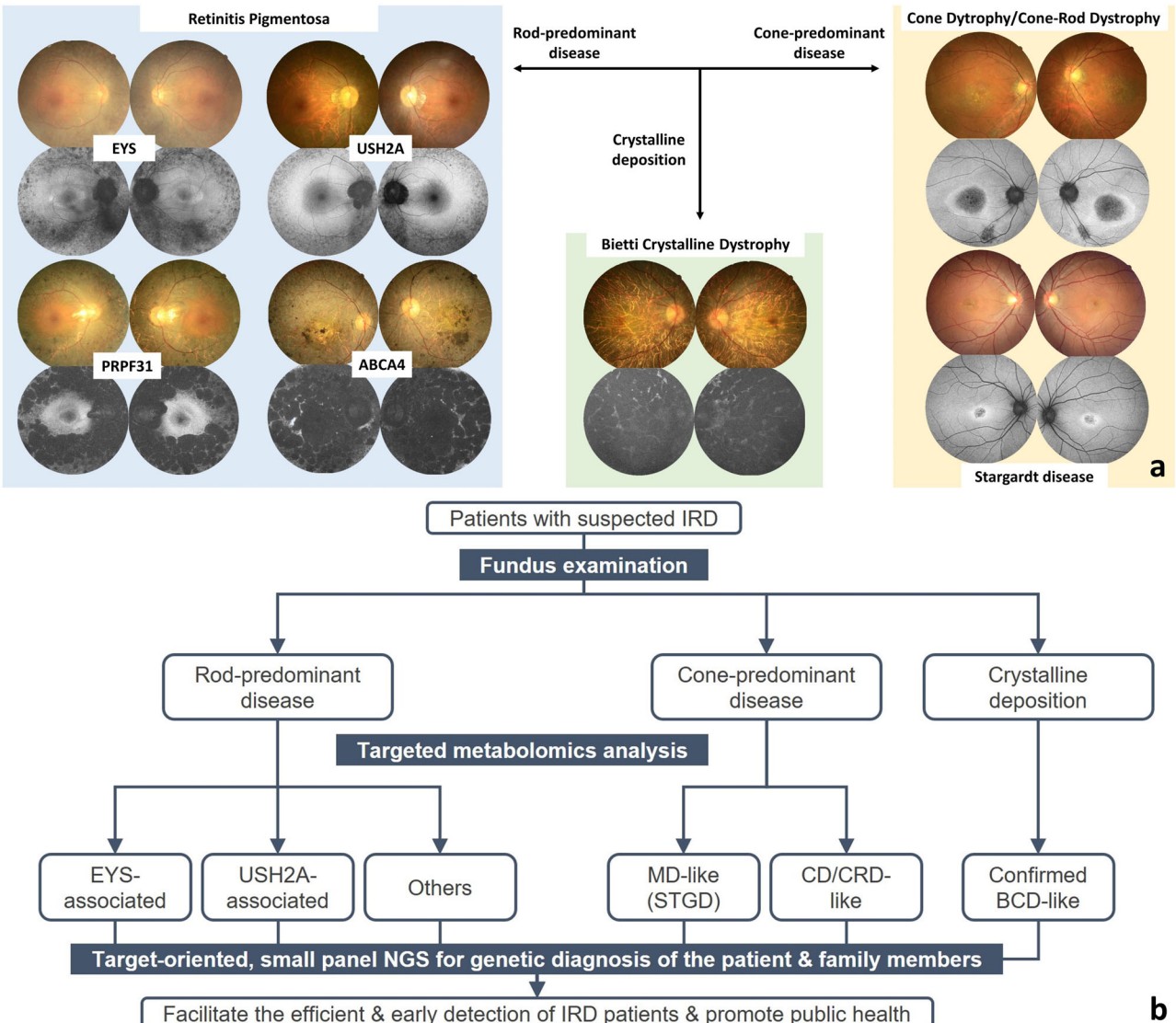

**Fig. 5 | Fundus photography and autofluorescence of representative cases and proposed diagnostic flow chart for IRDs. a** Different genotypes of RP cannot be differentiated by fundus examination, which shared common features, including pale disc, vessel attenuation, and pigmentary change, only with different disease severity. Moreover, the fundus appearance of CD/CRD and STGD is also undistinguishable, both with characteristic central maculopathy. However, the fundus appearance of BCD was characterized by crystalline deposition and could be differentiated easily from other IRDs. **b** we proposed a diagnostic flow chart to facilitate the early diagnosis of IRDs by incorporating clinical information, metabolomics analysis, and genetic diagnosis. IRDs belonging to rod-predominant disease, cone-predominant disease, and crystalline deposition were first determined by fundus examination. By incorporating targeted metabolomics analysis and a machine learning model, we could further differentiate *EYS* and *USH2A* genotypes in rod-predominant disease and STGD and CD/CRD in cone-predominant disease. Finally, we could reach a targeted, small panel NGS for the patient's and family members' genetic diagnosis. IRD, inherited retinal degeneration; BCD, Bietti's crystalline dystrophy, CD/CRD, cone dystrophy/cone-rod dystrophy, RP, retinitis; STGD, Stargardt disease, NGS, next-generation sequencing.

the field of imaging reading[41–44]. With the inspiring results of metabolomics analysis, we attempted to incorporate machine learning methods to enhance diagnostic ability in the present study. In cases with overlapping clinical presentations, such as STGD and CD/CRD, machine learning can be used to construct a diagnostic model with high accuracy. In addition, we demonstrated that the machine learning model could help differentiate between two common genotypes of RP, *EYS,* and *USH2A*, before reaching a relatively expensive genetic diagnosis. Machine learning enables us to manage massive amounts of data from serum metabolite analyses quickly and constitutes a diagnostic model.

Genetic augmentation therapy targeted specific genotypes of specific diseases have been developed as a potential treatment for IRDs in recent years[45]. Among numerous ongoing clinical trials in gene therapy for IRDs, Luxturna™ (Voretigene neparvovecryzl; Spark Therapeutics, Philadelphia, PA) is the first treatment approved by the United States Food and Drug Administration, targeting the RPE65-associated LCA cases[46]. Genetic therapy advancements have broadened the treatment of choice in IRDs, but the better prognosis often correlates with an earlier approach in cases with relatively preserved retinal function[47]. To launch the potential gene therapy for IRD cases in earlier disease stages, earlier and confirmative genetic and clinical diagnosis of IRD becomes even more important. Furthermore, besides the potential to reveal a diagnosis, metabolomic study has been applied for more clinical aspects in other common ophthalmological disease, including diabetes, age-related macular degeneration, glaucoma, and dry eye disease[48]. Specific metabolites identified from aqueous humor or serum are associated with disease activity,

progression, and also treatment response in patients with AMD and diabetic retinopathy[49–51]. Likewise, the metabolomic analysis in IRD could be expected to further correlate with disease prognosis, response to treatment, and discovering of available supplementation as well, with a larger study being conducted in the future. Evidence of metabolomic analysis in the diagnosis of IRDs was revealed in the present study. Incorporating metabolomic analysis helps facilitate differential diagnosis in patients suspected of having IRDs before acquiring genetic confirmation (Fig. 5b).

However, this study has some limitations. First, the number of cases was relatively small owing to the rare nature of IRDs. The influence of individual-specific and environmental factors on the metabolomics profile remains a pertinent consideration. In addition, as this was a cross-sectional study, we are not sure of the possible longitudinal changes in metabolomic expression in each individual or each IRD subtype. Future longitudinal studies are warranted to better understand the differences in metabolomes at different disease stages. Furthermore, expanding the participant pool is imperative to facilitate a more in-depth exploration of metabolomics within specific IRD genotypes and its correlation with disease severity and systemic characteristics. Absolute quantification for the identified metabolites would contribute to the establishment of a validated model applicable to new participants based on the findings of this study. Future studies should also focus on broadening the identification of metabolites with significant differences in and between these common IRDs, which could provide greater diagnostic value.

In this study, our results showed that serum metabolomic profiles differed significantly between healthy participants and those with IRD. The profiles also varied widely among the different IRD subtypes. By incorporating metabolomic analysis and fundus examination, we propose a diagnostic workflow for efficiently accessing IRDs. Our machine learning model could further identify the most common USH2A- and EYS-associated RP in rod-predominant diseases, as well as STGD and CD/CRD, which contributed the largest proportion of cone-predominant diseases. Our study provides a preliminarily exploratory overview of metabolomics in common IRDs and reveals the potential of metabolomic studies to enhance clinical diagnosis before conducting genetic consultation. Moreover, the information obtained from the metabolomic analysis could reflect genomic variations and possibly lead to future investigations into the pathophysiology and treatment of IRDs.

## Methods

### Study participants
Patients diagnosed with IRDs, including RP, STGD, CD/CRD, and BCD, at the National Taiwan University Hospital (NTUH) between 2015 and 2020 were prospectively enrolled in this cross-sectional observational study. The study protocol adhered to the tenets of the Declaration of Helsinki and was approved by the Institutional Review Board of NTUH. The diagnosis of IRD and identification of genotype were established by clinical data and NGS study targeting 212 IRD-related genes conducted in the TIP report[8]. It was based on a comprehensive ocular examination and was confirmed using panel-based NGS technology in every participant. A total of 28 healthy participants comprised the control group. Informed consent was obtained from all participants. Medical records and demographic data were recorded, including age, sex, body mass index, habit of smoking, supplementation of anti-oxidants, and history of systemic diseases requiring regular medication including hypertension, diabetes, cardiac disease, autoimmune disease, and cancer. Serum samples of all participants were collected within 1 to 3 p.m. without fasting status.

### Serum metabolites extraction
To extract metabolites from serum, the methyl tert-butyl ether (MTBE) extraction protocol was used with laboratory modification[52]. The serum (50 μL) was extracted by adding MTBE, methanol, and double-distilled water. The upper portion containing mostly lipids and the lower portion containing hydrophilic metabolites were separated, dried, and stored under −80 °C before analysis. The detailed extraction protocol is described in the Supporting Information.

### Liquid chromatography-mass spectrometry analysis
Liquid chromatography-mass spectrometry (LC-MS) analysis was performed using a Dionex UltiMate 3000 UHPLC system coupled with a Q Exactive Plus hybrid quadrupole-Orbitrap mass spectrometer (Thermo Fisher Scientific, USA). The samples (5 μL) were analyzed in random order. Pooled quality control samples were injected every 10 injections to ensure spectral quality. The upper (hydrophobic) and lower (hydrophilic) portions of the serum extract were separated using C18 (Waters UPLC CSH C18: 2.1 × 100 mm, 1.7 μm) and BEH amide columns (Waters UPLC BEH Amide: 2.1 × 150 mm, 1.7 μm), respectively. The detailed LC gradient settings are provided in the Supporting Information. Each sample was analyzed under both positive and negative ion modes. The mass range was set at m/z 150–1500 and 70–1000 for the analysis of the upper and lower portions, respectively. The detailed MS settings are listed in the Supporting Information.

### Statistical analysis and machine learning model construction
The LC-MS data were preprocessed using Thermo Scientific Compound Discoverer v3.2 software for peak alignment, background filtering, signal normalization, peak picking, and compound identification. The detailed settings for data preprocessing are described in the Supporting Information. Heatmap, volcano plot, and partial least squares-discriminant analysis (PLS-DA) were conducted using the MetaboAnalyst 5.0 online platform (https://www.metaboanalyst.ca/home.xhtml).

Two diagnostic models were constructed in this study: one for CD/CRD, STGD, and healthy participant classification and the other for genotype prediction in patients with RP. Diagnostic models were built based on the entire MS metabolic dataset using the commercially available data mining software RapidMiner Studio (version 9.10.001). First, the MS data were randomly separated into training and validation sets at a ratio of 7:3 for each subtype. The training dataset was used to train the Least Absolute Shrinkage and Selection Operator (LASSO) model and was evaluated using leave-one-out cross-validation. The trained model was evaluated using a validation dataset.

### Reporting summary
Further information on research design is available in the Nature Portfolio Reporting Summary linked to this article.

## Data availability
The data supporting the findings and for generating figures were provided in the Supplementary Information and also a Source Data file. The MS datasets have been deposited and are available at Metabolomics Workbench under the study ID ST003124 (https://doi.org/10.21228/M8PX4X). Raw data for clinical information are available from the corresponding author upon request. Source data are provided with this paper.

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

## Acknowledgements

The authors acknowledge the mass spectrometry technical research services from NTU Consortia of Key Technologies. This work was supported by the National Science and Technology Council, Taiwan, via the grants MOST 110-2314-B-002-225-MY3, MOST 111-2326-B-002-007-MY3, and MOST 112-2636-M-002-007-. We would like to thank Yun-Chen Hsieh for his assistance in conducting untargeted metabolomics analysis on serum samples.

## Author contributions

W.C. Wang 'contributed' methodology, analysis of data, and original draft preparation; C.H. Huang 'contributed' methodology, analysis of data, and original draft preparation; H.H. Chung 'contributed' methodology, and review and editing; P.L. Chen 'contributed' acquisition of data and analysis of data; F.R. Hu 'contributed' acquisition of data and analysis of data; C.H. Yang 'contributed' acquisition of data and analysis of data; C.M. Yang 'contributed' acquisition of data and analysis of data; C.W. Lin 'contributed' acquisition of data and analysis of data; C.C. Hsu 'contributed' conceptualization, methodology, interpretation of data, and review and editing; T.C. Chen 'contributed' conceptualization, methodology, interpretation of data, and review and editing. All authors have read and agreed to the published version of the manuscript.

## Competing interests

The authors declare no competing interests.

## Additional information

[1]Department of Chemistry, National Taiwan University, Taipei, Taiwan. [2]Department of Ophthalmology, Cathay General Hospital, Taipei, Taiwan. [3]School of Medicine, National Tsing Hua University, Hsinchu, Taiwan. [4]Graduate Institute of Medical Genomics and Proteomics, College of Medicine, National Taiwan University, Taipei, Taiwan. [5]Department of Medical Genetics, National Taiwan University Hospital, Taipei, Taiwan. [6]Department of Ophthalmology, National Taiwan University Hospital, Taipei, Taiwan. [7]Department of Ophthalmology, College of Medicine, National Taiwan University, Taipei, Taiwan. [8]Leeuwenhoek Laboratories Co. Ltd, Taipei, Taiwan. [9]Center of Frontier Medicine, National Taiwan University Hospital, Taipei, Taiwan. [10]Research Center for Developmental Biology and Regenerative Medicine, National Taiwan University, Taipei, Taiwan. [11]Department of Medical Research, National Taiwan University Hospital, Taipei, Taiwan. [12]These authors contributed equally: Wei-Chieh Wang, Chu-Hsuan Huang. [13]These authors jointly supervised this work: Cheng-Chih Hsu, Ta-Ching Chen. ✉e-mail: ccrhsu@ntu.edu.tw; tachingchen1@ntu.edu.tw

