## [Peer Review File · Nature Communications]

Reviewers' Comments:

Reviewer #1:

Remarks to the Author:

The authors analyzed serum metabolome in patients with IRDs and showed that serum metabolomic profiles differed between IRD patients and healthy controls. Also, they could differentiate EYS- and USH2A-RP using the machine learning model. This is an excellent work on metabolomics-based diagnostic approach in rare retinal diseases.

My two main concerns are as follows:

(1) The authors have not mentioned factors that could potentially affect metabolomic profiles in subjects. Many factors including fasting status at the time of serum collection, medications including antioxidants supplementation, smoking, systemic diseases, etc. can potentially affect metabolomic profiles. So, the authors need to describe those factors in included subjects and also need to discuss limitations of this study and reproducibility issue.

(2) Sample size issue: 70 patients with RP were included in this study. RP is a group of heterogenous retinal diseases, and the causative genes are over 100. Age, gender, disease status (i.e. early Vs advanced stage of RP), antioxidants supplementation, accompanied non-ocular diseases, etc. may affect the metabolomic profiles. So, I think 70 may not be good enough. How did the authors calculate sample size for this study? More extensive study using more samples and detailed information on patients' systemic characteristics may be necessary.

A few specific comments:

(1) In abstract, "EYS-, USH2A-associated, and other RP, being completely identical in clinical findings" is not correct, because the phenotypes of RP are not the same among the causative genes. Some genes have characteristic features that are distinguishable from RP by other genes.

(2) In discussion section, the authors mentioned that "Incorporating metabolomic analysis helps facilitate earlier diagnosis in patients suspected of having IRDs." I think the authors need to avoid too conclusive expression. We cannot conclude that metabolomic analysis is helpful for early diagnosis from this study.

(3) I recommend discussing further studies using metabolome analysis in IRDs. Metabolome analysis can be potentially useful for prediction of prognosis, disease status evaluation, response to treatment etc. as well as for diagnosis of IRDs.

Reviewer #3:

Remarks to the Author:

Summary: Using data from the Taiwan IRD Project (TIP) with 127 cases of inherited retinal degeneration (IRD) and 25 controls, the authors investigated the serum metabolomic profiles among patients of different IRD subtypes, There are two major findings: (1) the authors find that metabolites can distinguish different IRD subtypes and genotypes; and (2) the authors establish machine-learning lasso models to classify IRD subtypes prior to genetic testing.

Strengths of the study:

- The study addresses important questions in the field of IRD diagnosis and screening, and demonstrates the feasibility of using serum metabolites to facilitate early detection of potential IRD patients.
- The approaches are rigorous and well-described, from data processing, bioinformatic and statistical analysis for differential metabolite analysis, to machine learning lasso model construction for IRD subtype diagnosis and genotype prediction. The findings are well supported by the analyses conducted. The study limitations (e.g., small sample size and cross-sectional study design) are acknowledged and discussed.
- Given that genetic testing is time consuming but IRD gene therapy would be more effective if provided in early disease stages, the findings can facilitate early identification of IRD.

Specific comments:

- As the authors have noted in Supplementary Table 1, the age and sex distributions appear to be not similar across different IRD subtypes. It would be helpful to clarify if these factors (and/or any other confounders) were accounted for in the presented results (i.e., Fig 2 and Fig 3) regarding the association between IRD subtypes and metabolites. If possible confounders were not adjusted, it would be helpful to discuss the potential impact on the results, or to conduct sensitive analyses to understand the robustness of the findings.
- Related to the previous confounder adjustment--- how about the lasso models for predicting IRD subtypes and genotypes? What are the covariates besides metabolites included in the models?
- The lasso models were trained using leave-one-out cross-validation (CV), which is a good strategy given the moderate samples of the study. At the same time, it is also mentioned that the ROCs were computed on the entire (training + validation) data. I was wondering what the considerations are for using the entire data, if there could be an overfitting issue, and what the ROCs would look like if only based on validation data.
- A few technical detail questions regarding the machine learning model building: What are the criteria used to train the lasso model with the leave-one-out CV? What is "alpha=1" referring to? (Was it the alpha parameter in glmnet for specifying lasso and ridge?)

Reviewer #4:

Remarks to the Author:

This is a novel study to establish machine learning models to assist in diagnosis of retinal degeneration using results from serum metabolomics. They performed metabolomics on 28 healthy participants and 11-25 patients with different inherited retinal degenerations including BCD, CRD, STGD, ABCA4, PRPF31 EYE and USH2A. Except BCD, each patients had changes of distinctive metabolites, which were used to build training sets and validation sets.

Here are my concerns:

1. As mentioned in the limitations, the size for training set is too small. The validation sets used the samples that were analyzed from previous data sets to obtain the biomarkers to train. This may be the reason to have extremely high accuracy with such small training sizes. Ideally, these models can be used to test and predict the new patients.
2. The information for the patients needs more clarifications such as fed state or fasting state in sample collection, differences in their life style (diets, smoking, BMI), and differences in the time of the day in sample collection.
3. The lists of significantly changed metabolites or features should be included with parameters.
4. Are there overlapping changes among different IRDs?

Response to Reviewers

We thank the editor and reviewers for the comprehensive review and valuable comments, as well as for providing us with the opportunity to improve our manuscript. We have made revisions based on the suggestions and concerns. Please find our point-by-point responses below.

Reviewer's Comments:

Reviewer #1 (Remarks to the Author):

The authors analyzed serum metabolome in patients with IRDs and showed that serum metabolomic profiles differed between IRD patients and healthy controls. Also, they could differentiate EYS- and USH2A-RP using the machine learning model. This is an excellent work on metabolomics-based diagnostic approach in rare retinal diseases.

Answer:

We sincerely appreciate the reviewer's positive comments.

My two main concerns are as follows:

Comment 1.

(1) The authors have not mentioned factors that could potentially affect metabolomic profiles in subjects. Many factors including fasting status at the time of serum collection, medications including antioxidants supplementation, smoking, systemic diseases, etc. can potentially affect metabolomic profiles. So, the authors need to describe those factors in included subjects and also need to discuss limitations of this study and reproducibility issue.

Answer 1:

We appreciate the reviewer for the valuable comments. The time of serum collection for our patients was all united to 1 to 3 p.m. during scheduled appointment without fasting status for each case. Comorbidity of systemic disease that requiring regular medication was recorded, including hypertension, diabetes, cardiac disease, thyroid disease, autoimmune disease, and cancer. The proportion of prevalence of systemic disease in this study was between 16% to 31.3% in these groups and healthy controls, without statistical significance ($p=0.929$). Most IRD cases had regular anti-oxidant supplementation ($p=0.736$). There were only 3 cases had habit of smoking in this study (each in RP-USH2A, RP-EYS, and BCD group, $p=0.726$). However, we recognized that variable personal or environmental factors still possibly affect the individual metabolomics profile and we addressed this issue in the discussion about study limitation.

Therefore, we modified the manuscript as following, "Medical records and demographic data were recorded, including age, sex, body mass index (BMI), habit of smoking, supplementation of anti-oxidant, and history of systemic diseases requiring regular medication including hypertension, diabetes, cardiac disease, autoimmune disease, and cancer. Serum sample of all participants were collected within 1 to 3 p.m. without fasting status." in **1st paragraph of the method section**, and "There was no statistical difference between IRD groups and healthy control for presence of systemic disease, BMI, and habit of smoking ($p=0.929$, 0.895 , and 0.726 , respectively). The prevalence of anti-oxidant supplementation was similar in different IRD groups as well ($p=0.736$)." in **1st paragraph of the result section**, and "The influence of individual-specific and

environmental factors on the metabolomics profile remains a pertinent consideration.” in 8th paragraph of the discussion section. These data were also included in the modified **Supplementary Table 1**.

Comment 2.

(2) Sample size issue: 70 patients with RP were included in this study. RP is a group of heterogenous retinal diseases, and the causative genes are over 100. Age, gender, disease status (i.e. early Vs advanced stage of RP), antioxidants supplementation, accompanied non-ocular diseases, etc. may affect the metabolomic profiles. So, I think 70 may not be good enough. How did the authors calculate sample size for this study? More extensive study using more samples and detailed information on patients' systemic characteristics may be necessary.

Answer 2:

We appreciate the reviewer for the valuable comments. Since the low prevalence of inherited retinal degeneration (IRD) and retinitis pigmentosa (RP), the case number has been limited for this pioneer study in metabolomic analysis for IRD. As a rare disease, the prevalence of RP has been estimated as 1 in 3000 to 4000 people worldwide, and the incidence of IRD estimated from our national health insurance database was approximately 3 in 100,000 person-year. We enrolled samples of these 70 RP patients within a 5-year interval to achieve more than 20 patients with the leading genotypes, EYS and USH2A, and others, respectively. We agreed with the reviewer that further study with more participants is needed and could lead to more extensive investigations into the association with more detailed systemic characteristics

in specific RP-genotype. However, gathering enough samples from newly enrolled RP patients may require years to achieve due to the low disease prevalence. Expanding the participant pool would be our priority for further investigation in metabolomics in IRD.

In this presented study, we enrolled RP cases only with EYS, USH2A, ABCA4, and PRPF31 genotypes, which were four leading genotypes in our population in order to narrow genetic heterogeneity. The age and sex were similar within these genotypes as shown in supplementary table 1 ($p=0.935$ and 0.504 , respectively). There were 16%, 21.7%, 27.3%, and 18.2% of the enrolled cases having systemic disease, respectively in each genotype ($p=0.897$). Most of the cases (81.2% to 92%) had antioxidants supplementation ($p=0.316$). Accordingly, the major difference between was their genotypes and the metabolomics analysis could help to differentiate EYS and USH2A from other RP cases in combination with machine learning model.

Accordingly, we modified the manuscript in the **8th paragraph of the discussion section** as following, “Furthermore, expanding the participant pool is imperative to facilitate a more in-depth exploration of metabolomics within specific IRD genotypes and its correlation with disease severity and systemic characteristics.” We also added a short sentence “in this pioneer study” in the **9th paragraph of the discussion section** and modified **supplementary table 1** accordingly as well.

A few specific comments:

Comment 3.

(1) *In abstract, “EYS-, USH2A-associated, and other RP, being completely identical in clinical findings” is not correct, because the phenotypes of RP are*

not the same among the causative genes. Some genes have characteristic features that are distinguishable from RP by other genes.

Answer 3:

We appreciate the reviewer for the valuable comments. We agreed that certain phenotype-genotype correlations do exist in RP patients and could give us hints before genetic confirmation. We modified **the abstract** according to the recommendation as following, “EYS-, USH2A-associated, and other RP, sharing considerable similar characteristics in clinical findings”

Comment 4.

(2) In discussion section, the authors mentioned that “Incorporating metabolomic analysis helps facilitate earlier diagnosis in patients suspected of having IRDs.” I think the authors need to avoid too conclusive expression. We cannot conclude that metabolomic analysis is helpful for early diagnosis from this study.

Answer 4:

We appreciate the reviewer for the valuable comments. We modified the manuscript accordingly as following in **7th paragraph of the discussion section**, “Incorporating metabolomic analysis help to facilitate differential diagnosis in patients suspected of having IRDs before acquiring genetic confirmation.”

Comment 5.

(3) I recommend discussing further studies using metabolome analysis in IRDs.

Metabolome analysis can be potentially useful for prediction of prognosis, disease status evaluation, response to treatment etc. as well as for diagnosis of IRDs.

Answer 5:

We appreciate the reviewer's recommendation to broaden the discussion. Metabolome analysis has been applied in common ophthalmological disease including diabetes, age-related macular degeneration, glaucoma, etc. For example, metabolites associated in lipid metabolism pathway has been found to be involved in disease development in AMD, and disease progression and treatment response to anti-VEGF. Likewise, the metabolome analysis in IRD could be furtherly correlated with disease prognosis and response to available treatment as well, with larger study being conducted in the future. We modified the manuscript as following in the **7th paragraph of the discussion section**, "Furthermore, besides the potential in revealing diagnosis, metabolomic study has been applied for more clinical aspects in other common ophthalmological disease, including diabetes, age-related macular degeneration, glaucoma, and dry eye disease. Specific metabolites identified from aqueous humor or serum have been found to be associated with disease activity, progression, and also treatment response in patients with AMD and diabetic retinopathy. Likewise, the metabolomic analysis in IRD could be expected to furtherly correlate with disease prognosis, response to treatment, and discovering of available supplementation as well, with larger study being conducted in the future." Associated references were added to the reference list as order.

Reviewer #3 (Remarks to the Author):

Summary: Using data from the Taiwan IRD Project (TIP) with 127 cases of inherited retinal degeneration (IRD) and 25 controls, the authors investigated the serum metabolomic profiles among patients of different IRD subtypes, there are two major findings: (1) the authors find that metabolites can distinguish different IRD subtypes and genotypes; and (2) the authors establish machine-learning lasso models to classify IRD subtypes prior to genetic testing.

Strengths of the study:

- The study addresses important questions in the field of IRD diagnosis and screening, and demonstrates the feasibility of using serum metabolites to facilitate early detection of potential IRD patients.*
- The approaches are rigorous and well-described, from data processing, bioinformatic and statistical analysis for differential metabolite analysis, to machine learning lasso model construction for IRD subtype diagnosis and genotype prediction. The findings are well supported by the analyses conducted. The study limitations (e.g., small sample size and cross-sectional study design) are acknowledged and discussed.*
- Given that genetic testing is time consuming but IRD gene therapy would be more effective if provided in early disease stages, the findings can facilitate early identification of IRD.*

Answer:

We sincerely appreciate the reviewer's positive comments.

Specific comments:

Comment 1.

• *As the authors have noted in Supplementary Table 1, the age and sex distributions appear to be not similar across different IRD subtypes. It would be helpful to clarify if these factors (and/or any other confounders) were accounted for in the presented results (i.e., Fig 2 and Fig 3) regarding the association between IRD subtypes and metabolites. If possible confounders were not adjusted, it would be helpful to discuss the potential impact on the results, or to conduct sensitive analyses to understand the robustness of the findings.*

Answer 1:

We appreciate the reviewer for the valuable comments. The sex distribution was not even between IRD groups but without statistical significance ($p=0.352$). Age distribution was only significant for the cases with Stargardt's disease (STGD), which was younger than healthy participants and cases with RP and BCD. We applied analysis of covariance (ANCOVA) to adjust the confounder "age" to the levels of specific identified metabolites in Figure 2 and the Volcano plots in Figure 3. The results were much similar with the original ones that BCD and healthy participants shared similar metabolomic profiles and were different from other IRD groups. There still weren't any metabolite feature could be identified between BCD and healthy participants in Volcano plot, and even fewer metabolite features were identified between STGD and RP-ABCA4 cases, when adjusted with age. We added these data into **Supplementary Figure 1 and 2** to report these findings after adjustment of the age confounder.

Comment 2.

• *Related to the previous confounder adjustment--- how about the lasso models for predicting IRD subtypes and genotypes? What are the covariates besides metabolites included in the models?*

Answer 2:

We appreciate the reviewer for the valuable comments. In our machine learning models, age and sex has not been selected as covariates. There was no statistical difference in sex distribution in our study groups and normal subjects. The age in STGD group is significantly younger than healthy participants, RP, and BCD groups, but not significant with CD/CRD group. The first machine learning model for differentiating STGD, CD/CRD cases from normal subjects could achieve 100% accuracy in validation set. The accuracy remained the same when adding age and sex as covariates and the weighting of this covariate (age) was only 0.5, while sex has not been selected by the model. The accuracy in validation set decreased from 85.7% to 76.2% in the second machine learning model for differentiating EYS, USH2A genotypes from other RP when adding age and sex as covariates. Therefore, we reported the two machine learning models with only metabolites being included as covariates. We added the list of metabolites selected by machine learning models and the corresponding weighting in **Supplementary table 3**.

Comment 3.

• *The lasso models were trained using leave-one-out cross-validation (CV), which is a good strategy given the moderate samples of the study. At the same time, it is also mentioned that the ROCs were computed on the entire (training*

+ validation) data. I was wondering what the considerations are for using the entire data, if there could be an overfitting issue, and what the ROCs would look like if only based on validation data.

Answer 3:

We appreciate the reviewer for the valuable comments. We added ROCs based on the validation data and modified figure 4 accordingly.

Comment 4.

• A few technical detail questions regarding the machine learning model building: What are the criteria used to train the lasso model with the leave-one-out CV? What is “alpha=1” referring to? (Was it the alpha parameter in glmnet for specifying lasso and ridge?)

Answer 4:

We appreciate the reviewer for the valuable comments. The machine learning model was built using commercially available software RapidMiner Studio (version 9.10.001). To implement the lasso model, we used the built-in generalized linear model and set the alpha value to 1. The alpha value is the same as the alpha parameter in glmnet which specify lasso (alpha=1) and ridge (alpha=0). The lambda parameter, which controls the amount of regularization, was screened using the “lambda search” function in the RapidMiner. The optimization process stops when the relative improvement is lower than 0.001.

We have elaborated more on the details of machine learning model building in the supporting information as following, “In this study, RapidMiner Studio (version 9.10.001), a commercially available data mining software, was

used to construct the machine learning models. To implement the lasso model, the built-in generalized linear model was used and the alpha parameter was set to 1, indicating the use of an L1 penalty. The lambda parameter, controlling the degree of regularization, was determined using the “lambda search” function within RapidMiner. The optimization process programmed to terminate when the relative improvement fell below 0.001.”

Reviewer #4 (Remarks to the Author):

This is a novel study to establish machine learning models to assist in diagnosis of retinal degeneration using results from serum metabolomics. They performed metabolomics on 28 healthy participants and 11-25 patients with different inherited retinal degenerations including BCD, CRD, STGD, ABCA4, PRPF31 EYE and USH2A. Except BCD, each patient had changes of distinctive metabolites, which were used to build training sets and validation sets.

Answer:

We sincerely appreciate the reviewer’s positive comments.

Here are my concerns:

Comment 1.

1. As mentioned in the limitations, the size for training set is too small. The validation sets used the samples that were analyzed from previous data sets to obtain the biomarkers to train. This may be the reason to have extremely high accuracy with such small training sizes. Ideally, these models can be used to test and predict the new patients.

Answer 1:

We appreciate the reviewer for the valuable comments. Because of small sample size, we used leave-one-out cross-validation for evaluating training set. The training set and validation set were randomly selected before applying machine learning model. Therefore, all the samples in the validation set hadn't been used during model training process.

The patients in the presented study groups were enrolled within a 5-year interval. The prevalence of IRD was approximately 1 in 2500 people and many of them belonged to diagnosis of RP. The incidence of IRD estimated from our national health insurance database was approximately 3 in 100,000 person-year. Therefore, gathering enough samples from newly enrolled IRD patients in every group as new validation set is time-demanding due to their rare-disease nature. Absolute quantification and standardization for specific metabolites that identified in the diagnostic models should be performed in the future. We would focus on enrollment of new participant to validate the result subsequently. Further study would be conducted based on the findings from this first investigation of metabolomic profiles in IRD patients.

We modified the manuscript as following in the **8th paragraph of discussion section**, “Furthermore, expanding the participant pool is imperative to facilitate a more in-depth exploration of metabolomics within specific IRD genotypes and its correlation with disease severity and systemic characteristics. Absolute quantification for the identified metabolites would contribute to the establishment of a validated model applicable to new participants based on the findings of this study.”

Comment 2.

2. The information for the patients needs more clarifications such as fed state or fasting state in sample collection, differences in their life style (diets, smoking, BMI), and differences in the time of the day in sample collection.

Answer 2:

We appreciate the reviewer for the valuable comments. The time of serum collection was united to 1 to 3 p.m. during scheduled appointment without fasting status for all cases. There were only 3 cases had habit of smoking in this study (each in RP-USH2A, RP-EYS, and BCD group, $p=0.726$). The BMI was similar between different groups around 22.0 to 23.3 ($p=0.895$). We modified the manuscript as following, "Medical records and demographic data were recorded, including age, sex, body mass index (BMI), habit of smoking, supplementation of anti-oxidant, and history of systemic diseases requiring regular medication including hypertension, diabetes, cardiac disease, autoimmune disease, and cancer. Serum sample of all participants were collected within 1 to 3 p.m. without fasting status." in **1st paragraph of the Method section**, and "There was no statistical difference between IRD groups and healthy control for presence of systemic disease, BMI, and habit of smoking ($p=0.929$, 0.895 , and 0.726 , respectively). The prevalence of anti-oxidant supplementation was similar in different IRD groups as well ($p=0.736$)." in the **1st paragraph of the Result section**. These data were also included in the modified **Supplementary table 1**.

Comment 3.

3. The lists of significantly changed metabolites or features should be included

with parameters.

Answer 3:

We appreciate the reviewer for the valuable comments. We added the weighting parameters to the list of metabolites selected by machine learning models in **Supplementary table 3**.

Comment 4.

4. Are there overlapping changes among different IRDs?

Answer 4:

We appreciate the reviewer for the valuable comments. In metabolomics analysis, we found that cases with BCD had more similar profile with healthy subjects and different from other IRD. Specific metabolites had been identified following this trend, that elevated D-xylonate, citronellyl acetate, and hexadecanedioic acid levels were observed in all IRD subtypes, except BCD, compared with that of the control group (**Figures 2b-2d**). In contrast, decreased concentrations of N-undecanoylglycine and the other three glycerophospholipids, phosphatidylserine (14:1/16:0), phosphatidylcholine [16:0/9:0(CHO)], and phosphatidylcholine (19:1), were found in all IRD subtypes, except BCD, compared with that of the control group (**Figures 2e-2h**). (*Information above were mentioned in the 2nd paragraph of the Result section of revised manuscript*). BCD is caused by mutations in the CYP4V2 gene, which is responsible for the transformation of polyunsaturated fatty acids. The pathogenesis of BCD is dysfunction of lipid metabolism and is much different from other subtypes of IRD, which generally affect enzymes involved

in the visual pathway or protein, maintaining structural stability in photoreceptors and retinal pigmented epithelium cells. (*Information above were mentioned in the 4th paragraph of the Discussion section of revised manuscript*).

Once again, we wish to express our great appreciation to the editors and reviewers for their valuable comments and for providing us with the opportunity to revise our paper. We hope the revised manuscript is suitable for publication in the journal.

Sincerely,

Cheng-Chih Hsu, Ph.D

Corresponding author

No. 1, Sec. 4, Roosevelt Rd., Taipei, Taiwan

Telephone Number: 886-2-3366-3844

E-mail: ccrhsu@ntu.edu.tw

Ta-Ching Chen, MD, Ph.D

Corresponding author

No 7, Chung-Shan S. Rd., Taipei, 100, Taiwan.

Office voice: 886-2-23123456 ext 63783

E-mail: Tachingchen1@ntu.edu.tw

Reviewers' Comments:

Reviewer #1:

Remarks to the Author:

The authors responded well to the reviewers' comments.

Reviewer #4:

Remarks to the Author:

The authors addressed all my comments.

Reviewer #5:

Remarks to the Author:

This work presents a pioneer work for the diagnosis of common inherited retinal degenerations based on metabolomics data. The study seems sound and innovative. Authors have correctly addressed the comments of the previous round of review. However, I have some methodological questions that need clarification.

1) How many metabolites were used in the Lasso regression model? Did authors conduct any feature selection based on ANOVA or just include all the metabolites detected by mass spectrometry? If the authors carried out feature selection, this cannot be done with the whole set of data. Only training data can be used for feature selection. This is not sufficiently clear in the text.

2) It is not clear in the Volcano plot whether p-values were correctly for multiple hypothesis testing. This is not detailed neither in the text nor in the legend of Figure 3.

3) Authors should compare the impact of metabolomics data in the predictive capacity of the Lasso model when mixed with other genetic and clinical variables. In other words, authors should include: a) clinical-genetic Lasso model; b) clinical-genetic-metabolomic Lasso model; c) metabolomic Lasso model.

4) In my opinion, Lasso regression is more to do with statistical modeling than a machine learning. Why not to consider machine learning models, such as Support Vector Machines, Classification Trees, Random Forest, among others? Have authors compared the performance of different methods?

Response to Reviewers

We sincerely thank the Editor and Reviewers for the comprehensive review and valuable comments, as well as for providing us with the opportunity to improve our manuscript. We have made revisions based on the suggestions and concerns. Please find our point-by-point responses below.

Reviewer's Comments:

Reviewer #1 (Remarks to the Author):

The authors responded well to the reviewers' comments.

Answer:

We sincerely appreciate the reviewer's positive comments.

Reviewer #4 (Remarks to the Author):

The authors addressed all my comments.

Answer:

We sincerely appreciate the reviewer's positive comments.

Reviewer #5 (Remarks to the Author):

This works presents a pioneer work for the diagnosis of common inherited retinal degenerations based on metabolomics data. The study seems sound and innovative. Authors have correctly addressed the comments of the previous round of review. However, I have some methodological questions that need clarification.

Answer:

We sincerely appreciate the reviewer's positive comments and please find our point-by-point responses below:

Specific comments:

Comment 1.

• *How many metabolites were used in the Lasso regression model? Did authors conduct any feature selection based on ANOVA or just include all the metabolites detected by mass spectrometry? If the authors carried out feature selection, this cannot be done with the whole set of data. Only training data can be used for feature selection. This is not sufficiently clear in the text.*

Answer 1:

We appreciate the reviewer for the valuable comments. In our methodology, we didn't conduct feature selection through ANOVA during machine learning model construction. We adopted a comprehensive approach wherein all 1,606 metabolites detected by mass spectrometer were included in the initial dataset.

Specifically, for the development of the diagnostic model targeting cone-rod dystrophy/Stargardt disease/normal samples classification, and RP genotyping, we employed the Lasso regression algorithm. Through this process, we identified 5 crucial metabolites for the former and 14 for the latter as the key diagnostic features from the mass spectrometry metabolic dataset.

We modified the manuscript accordingly as follows **in the 5th paragraph of the Methods section:** “Diagnostic models were built based on entire MS metabolic dataset using the commercially available data mining software RapidMiner Studio (version 9.10.001).”

Comment 2.

• *It is not clear in the Volcano plot whether p-values were corrected for multiple hypothesis testing. This is not detailed neither in the text nor in the legend of*

Figure 3.

Answer 2:

We appreciate the reviewer for the valuable comments. Yes, we have conducted multiple hypothesis testing for the *p-values* shown in the Volcano plot.

We modified the manuscript accordingly as follows **in the 5th paragraph of the Results section:** “The criteria for significant features highlighted in plots were defined as a false discovery rate < 0.05 based on Benjamini–Hochberg test and fold change > 2.”

The legend of **Figure 3.** has also been modified as: “The significant features highlighted in the volcano plot were defined as a false discovery rate < 0.05 (Benjamini–Hochberg test) and fold change > 2.”

Comment 3.

• *Authors should compare the impact of metabolomics data in the predictive capacity of the Lasso model when mixed with other genetic and clinical variables. In other words, authors should include: a) clinical-genetic Lasso model; b) clinical-genetic-metabolomic Lasso model; c) metabolomic Lasso model.*

Answer 3:

We appreciate the reviewer for the valuable comments. Yes, inherited retinal degeneration (IRD) is a group of diseases with high heterogeneity in both phenotype and genotype. Similar clinical presentation could often be found in patients with different disease entity and also diverse genetic background. As described the Figure 5B, patients with subjectively blurred vision may seek medical advices, and clinical presentation, such as fundus finding and

electroretinogram, would be firstly defined and categorized. However, some patients classified in the same clinical category (for example, rod-predominant, cone-predominant, and so on) may have different genetic defects and different pathophysiologic change. To achieve pathological classification and to engage the novel therapies in near future, we hope to explore more molecular-based diagnoses for individual. Therefore, in this presented study, we aimed to incorporate metabolomic analysis and machine learning model to facilitate diagnostic workflow for these IRDs with overlapping clinical presentation, constructing a bridge to genetic confirmation, which is currently the golden standard for the diagnosis of IRD. Therefore, we focused on reporting the Lasso models with metabolomics features only to facilitated diagnosis and we are happy that both these two metabolomic models (within rod-predominant cases and within cone-predominant cases) yielded satisfactory results. We thank the reviewer for the important comments again.

Comment 4.

• In my opinion, Lasso regression is more to do with statistical modeling than a machine learning. Why not to consider machine learning models, such as Support Vector Machines, Classification Trees, Random Forest, among others? Have authors compared the performance of different methods?

Answer 4:

We appreciate the reviewers' insightful comments regarding the choice of Lasso regression in our study and the suggestion to consider alternative machine learning models. The selection of Lasso regression was driven by its dual capability for variable selection and regularization, addressing challenges in high-dimensional datasets like our metabolomic data.

While rooted in statistical modeling, Lasso regression has gained prominence in machine learning due to its effectiveness in managing multicollinearity and preventing overfitting. Its widespread use in metabolomic-based diagnostic applications, such as cardiometabolic risk assignment¹, cancer classification^{2,3}, and COVID-19 diagnosis⁴. Our prior utilization of the Lasso regression in metabolomics data analysis has demonstrated its efficacy in accurately diagnosing diseases such as breast cancer⁵ and parasitic infections⁶, thereby emphasizing its robust performance. Moreover, the Lasso regression plays a crucial role in enhancing data interpretation, further solidifying its versatility and utility in diverse contexts.

To identify the most suitable model to predict different IRD subtypes, at the beginning of the study, we initiated a thorough comparative analysis. This analysis compares the predicting accuracy of various models, including Lasso, Classification Trees, Random Forest, and Deep Learning. The result showcases the superior performance of the Lasso regression (Table 1), which validates our deliberate selection.

Table 1. The performance of different machine-learning models for IRD subtypes classification. Two diagnostic models were established to differentiate (1) CD/CRD, STGD, and control group and (2) RP with *EYS*, *USH2A*, and other genotypes, using the machine learning LASSO, classification tree, random forest, and deep learning model. The MS data were randomly separated into training and validation sets with a ratio of 7 to 3 for each subtype. The machine learning models were trained using the training dataset and evaluated using leave-one-out cross-validation.

Model	Data set	LASSO	Classification tree	Random forest	Deep learning
CD/CRD, STGD, and Control	Training set	100% +/- 0.0%	91.8% +/- 27.7%	100% +/- 0.0%	90.0% +/- 10.5%
RP genotyping	Training set	84.0% +/- 15.8%	90.0% +/- 30.3%	72.0% +/- 45.4%	66.0% +/- 47.9%

Reference:

1. Beyene, H. B., Giles, C., Huynh, K., Wang, T., Cinel, M., Mellett, N. A., ... & Meikle, P. J. (2023). Metabolic phenotyping of BMI to characterize cardiometabolic risk: evidence from large population-based cohorts. *Nature Communications*, 14(1), 6280.
2. Sans, M., Zhang, J., Lin, J. Q., Feider, C. L., Giese, N., Breen, M. T., ... & Eberlin, L. S. (2019). Performance of the MasSpec Pen for rapid diagnosis of ovarian cancer. *Clinical chemistry*, 65(5), 674-683.
3. Zhang, J., Rector, J., Lin, J. Q., Young, J. H., Sans, M., Katta, N., ... & Eberlin, L. S. (2017). Nondestructive tissue analysis for ex vivo and in vivo cancer diagnosis using a handheld mass spectrometry system. *Science translational medicine*, 9(406), eaan3968.
4. Garza, K. Y., Silva, A. A. R., Rosa, J. R., Keating, M. F., Povilaitis, S. C., Spradlin, M., ... & Porcari, A. M. (2021). Rapid screening of COVID-19 directly from clinical nasopharyngeal swabs using the MasSpec Pen. *Analytical chemistry*, 93(37), 12582-12593.
5. Huang, Y. C., Chung, H. H., Dutkiewicz, E. P., Chen, C. L., Hsieh, H. Y., Chen, B. R., ... & Hsu, C. C. (2019). Predicting breast cancer by paper spray ion mobility spectrometry mass spectrometry and machine learning. *Analytical chemistry*, 92(2), 1653-1657.
6. Wang, W. C., Chung, H. H., Dutkiewicz, E. P., Wong, J. Y., Yang, W. C., Chang, C. L. T., & Hsu, C. C. (2021). On-site Diagnosis of Poultry Coccidiosis by a Miniature Mass Spectrometer and Machine Learning. *ACS Agricultural Science & Technology*, 2(1), 17-21.

Once again, we wish to express our great appreciation to the editors and reviewers for their valuable comments and for providing us with the opportunity to revise our paper. We hope the revised manuscript is suitable for publication in the journal.

Sincerely,

Cheng-Chih Hsu, Ph.D

Corresponding author

No. 1, Sec. 4, Roosevelt Rd., Taipei, Taiwan

Telephone Number: 886-2-3366-3844

E-mail: ccrhsu@ntu.edu.tw

Ta-Ching Chen, MD, Ph.D

Corresponding author

No 7, Chung-Shan S. Rd., Taipei, 100, Taiwan.

Office voice: 886-2-23123456 ext 63783

E-mail: Tachingchen1@ntu.edu.tw

Reviewers' Comments:

Reviewer #5:

Remarks to the Author:

Authors have convincingly addressed all my previous comments.